# FG-Attn: Leveraging Fine-Grained Sparsity In Diffusion Transformers

## Abstract

Generating realistic videos/images with diffusion transformers requires evaluating attention over extremely long sequences, with attention layers accounting for the majority of generation latency. Exploiting sparsity in attention maps offers a promising opportunity to reduce this cost. However, existing methods rely on block-sparse attention, which skips attention computation only when all scores within a coarse M×M tile (typically 64×64) are expected to be negligible. This coarse-grained skipping leaves a large fraction of redundant computation unaddressed. In this work, we show that attention maps in diffusion transformers exhibit significant *fine-grained* sparsity.

Leveraging this efficiently on modern GPUs is challenging, as fine-grained skipping introduces irregular memory access, can reduce tensor core utilization, and it is difficult to determine which computation to skip without loss in accuracy. We propose FG-Attn, a novel fine-grain sparse attention mechanism that skips score computations at the granularity of M×1 slices, where each slice is the result of query-key dot products between M query vectors and a *single key*. We introduce a highly efficient asynchronous gather-load primitive that loads only the sparse set of key/value vectors into tensor-core-compatible tiles in the on-chip GPU shared memory, hiding the overhead of irregular memory access. We develop two training-free, lightweight prediction strategies that identify redundant scores to skip with negligible overhead. FG-Attn can fully supercede existing block sparsity methods in DiTs, and we demonstrate that it achieves up to $1.65\times$ speedup ($1.48\times$ on avg) for state-of-art video models on an H100 GPU.

## 1 Introduction

Media generation models in deep learning have proven highly effective at capturing complex data distributions across multiple modalities, including videos (Wan et al., 2025; Kong et al., 2024), audio (Kong et al., 2020), 3D models (Zhao et al., 2025) and images (Esser et al., 2024b;a; Chen et al., 2023). When trained on large-scale datasets, these models can synthesize realistic, high-quality content, enabling transformative applications such as advanced video editing, intuitive 3D modeling, and immersive environments. These models are powered by diffusion models, a type of generative deep learning models that generate synthetic data by iteratively refining (or *denoising*) a latent space representation of the data using a *denoising function*. In video generation, each video is represented as a long sequence of embedding vectors that are progressively refined by a diffusion transformer (DiT) (Peebles & Xie, 2023) before being decoded into video frames.

Even short, low-resolution videos yield extremely long embedding sequences. For example, a state-of-art video DiT model, the Wan 2.1 1.3B (Wan et al., 2025), encodes a 5-second video at 720p resolution into 74000 embedding vectors, requiring 5 minutes for video generation on an H100 GPU (the Wan 2.1 14B model requires over 25 minutes). The majority of this generation latency is incurred by the attention layers of the transformer - $91\%$ of the runtime for this example with Wan 2.1. Since attention scales quadratically with sequence length, higher resolution or longer videos rapidly exacerbate latencies: generating a 10-second video at 720p requires twice as many embeddings, resulting in $\sim 4x$ the generation latency. Thus, the latency becomes increasingly dominated by attention computation. For Wan 2.1 1.3B, attention layers take $76\%$ of the overall latency to produce 49 frames and $91\%$ for 81 frames.

It is well-known that the input to DiTs contains significant redundancy. Prior work show that many query–key pairs yield negligible attention scores, and computing only a fraction of the full matrix still produces perceptually valid videos (Xu et al., 2025; Li et al., 2025; Xi et al., 2025). As shown in Fig.1, computing only 20% of the attention scores per head still produces a valid video with no noticeable loss in quality. Note that the generated video frames are slightly different because the denoising process is sensitive to small perturbations in early denoising iterations, but the diffusion model still captures the same video distribution.

Several works exploit this redundancy to accelerate video generation: RadialAttention (Li et al., 2025), X-Attention (Xu et al., 2025), and SparseVideoGen (Xi et al., 2025) observe that important attention scores are concentrated in certain regions of the map, particularly near the diagonal, and use static masks to skip redundant computation. However, since these methods skip a fixed subset of scores at each head, they may also omit essential ones. To address this, Video Sparse Attention (VSA) (Zhang et al., 2025e) instead infers the mask dynamically at runtime using additional parameters. These approaches typically rely on block sparse attention (section 3.2) as the underlying sparse attention mechanism. Block sparse attention mechanisms (Guo et al., 2024; Ye et al., 2025; Hong et al., 2023; Dao et al., 2022; Wang et al., 2024; Dong et al., 2024) divide the attention score matrix into coarse tiles of size M×M (typically M=64) aligned to GPU tensor-core dimensions, for $M$ query tokens and $M$ key tokens. A full tile of query–key dot products is skipped only if *all scores* within it are predicted to be near zero.

We however observe that attention maps in diffusion transformers exhibit *fine-grained sparsity*: many query–key products are near-zero even when others in the same block are not. Exploiting this finer structure has the potential to substantially reduce FLOPs. For example, skipping $16 \times 16$ blocks of attention can reduce the required FLOPs by up to 70%, compared to only $\sim 15\%$ with $64 \times 64$ block sparsity, without noticeable degradation in video quality (see section 3.2). Similarly, skipping $128 \times 1$ slices of attention scores can reduce FLOPs by as much as $55\%$. These observations demonstrate that leveraging fine-grained sparsity in attention maps can unlock significantly larger reductions in computation than coarse-grained block-sparse methods allow.

Our goal in this work is twofold: (1) to design a highly efficient sparse attention mechanism that can skip attention score computations at a finer granularity than block-sparse attention, and (2) to introduce a lightweight method for predicting which fine-grain subsets of attention scores can be safely skipped, i.e, constructing the sparse attention mask. To this end, we introduce FG-Attn, an efficient fine-grained sparse attention mechanism for diffusion transformers that skips computing scores corresponding to

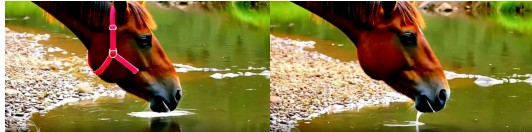

Figure 1: Left: A video frame from Wan 2.1 (Wan et al., 2025) for the prompt "horse bending and drinking water from a lake". Right: The same model generates a similar video using only 20% of attention scores per head, achieving comparable results with a fraction of the FLOPs.

*slices* of size $M \times 1$ of the attention map, i.e., attention scores produced by one key and a group of $M$ contiguous queries. Implementing fine-grained sparse attention in modern GPUs is non-trivial: (i) Fine-grained masks destroy the regular tiling structure expected by GPU tensor-core kernels, leading to irregular memory access patterns that prevent coalesced loads. Relevant key/value vectors must be gathered from non-contiguous locations, which can dominate runtime if handled naively. (ii) Because tensor cores operate on fixed-size dense tiles, unstructured sparsity can leave much of the hardware underutilized. (iii) Determining the sparse mask at a fine granularity requires reliably and efficiently predicting the importance of individual M×1 slices without explicitly computing them.

For (i) and (ii), the kernel must fetch only the sparse subset of relevant key/value vectors for a group of queries from high-bandwidth memory (HBM) and assemble them into the on-chip GPU shared memory. in the exact tile format expected by tensor cores. We address this by introducing a new asynchronous gather-load primitive, which gathers non-contiguous key/value vectors and repacks them into swizzled tiles in shared memory. Because current GPUs lack hardware support such as a Tensor Memory Accelerator (TMA) for indirect, sparse address generation, we emulate this functionality using existing asynchronous load instructions. This address generation and data packing is overlapped with attention computation to effectively hide the latency of irregular memory access.

For (ii), we propose two training-free strategies to identify the set of keys to load for a group of queries. The first draws inspiration from caching techniques in diffusion inference: within each at-

tention head, query-key pairs with significant scores remain stable across denoising iterations. Thus, in the first iteration we compute the full attention map, identify significant slices via thresholding, and reuse this mask in later steps. The main limitation is memory overhead from storing cached masks. To mitigate this, we introduce a second lightweight strategy: for queries $q_1, q_2, \ldots, q_M$, we compute dot products only for keys likely to yield a significant score with the mean query $q_{\text{mean}}$. Instead of evaluating all keys, we load only the top-$k$ by score with $q_{\text{mean}}$ (section 4.3). This heuristic is motivated by the observation that nearby embeddings often produce similar query distributions, allowing accurate approximation with far fewer key computations.

We demonstrate that FG-Attn enables faster video and image generation without sacrificing the output quality. On state-of-art generation models, we show that FG-Attn speeds up the video generation time by up to $1.65\times$ ($1.48\times$ on average). Our contributions are:

- We demonstrate that video diffusion models contain a significant amount of fine-grain sparsity in their attention maps that are not leveraged by existing block sparse attention methods.
- We introduce the first slice-based sparse attention mechanism that can practically exploit fine-grained sparsity on modern GPUs. To support this, we design a novel asynchronous gather-load primitive, which efficiently assembles sparse key/value vectors into tensor-core-compatible tiles, overcoming the overheads of irregular memory access.
- We demonstrate that sparsity patterns remain stable across denoising iterations, enabling a cache-based thresholding strategy that avoids recomputation while preserving accuracy.
- We propose two lightweight strategies for sparse mask generation that operate entirely without retraining. This ensures FG-Attn is directly applicable to existing state-of-art video DiTs.
- We show that our sliced attention mechanism can fully supersede existing block-sparse attention methods in DiTs, achieving performance equivalent to or better than all prior coarse-grained approaches with negligible accuracy loss.

## 2 BACKGROUND

### 2.1 VIDEO DIFFUSION TRANSFORMER MODELS

Diffusion models are a class of deep learning models that fit the gradient of the $\log$ probability (score function) of the data distribution $x_d$, i.e., $s(x_d) = \nabla_x \log(p_{\text{data}})$ using a parameterized model. Latent space diffusion models fit the score function of the latent space representation of the data, denoted by $x$. The latent space representation consists of embedding vectors that can be decoded to or encoded from a data sample using an autoencoder. Fig. 2a shows how video can be encoded into a sequence of embedding vectors using a variational autoencoder model, which are flattened to form a single sequence of embeddings. Producing a video corresponds to drawing a sample from the probability flow ODE $dx = s(x)dt$, where $s(x) = \nabla \log(p_{\text{data}}(x))$ is the fitted score function. This ODE is solved numerically using integrators such as DPM-Solver (Lu et al., 2022), which iteratively evaluate the score function to update the latent representation $x$. This iterative refinement, known as *denoising*, generates videos by starting from a noisy vector that a transformer-based denoising function progressively refines into into embeddings for clear video frames (Fig. 2b).

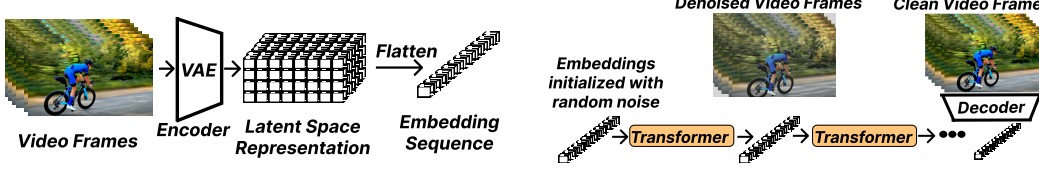

(a) A set of noisy video frames are encoded into a sequence of latent space embeddings using a VQVAE.

(b) The diffusion transformer denoises the embeddings to generate clean video frame embeddings.

Figure 2: Latent space representation of video frames, represented as a set sequence of embeddings and encoded using a VQVAE, can be denoised using a DiT to produce embeddings of clean frames.

### 2.2 GPU ARCHITECTURE BACKGROUND

A GPU consists of high-bandwidth memory (HBM) for global storage and an array of streaming multiprocessors (SMs) for computation. Each SM provides shared memory (a fast on-chip scratchpad) and tensor cores for matrix operations. Efficient execution requires first staging tensor data

from HBM into shared memory, since directly operating from HBM is too slow. Once in shared memory, tensor cores can process data while subsequent transfers are overlapped with computation. Parallel programs are mapped to GPUs by dividing work into thread blocks. Each thread block contains many threads, grouped into warps of 32. On Hopper-class GPUs (e.g., H100), four warps form a warp group that can issue a matrix multiply–accumulate (MMA) instruction to efficiently utilize a tensor core. (Figure and more details in Appendix B.)

**Writing efficient GPU kernels.** Efficient kernels keep the tensor cores continuously saturated by overlapping data movement with computation. Best practices of achieving this is to divide threads in a block into producers and consumers (at the warp group granularity in Hopper). Producer warp groups load tensor data from HBM into shared memory, while consumer warp groups issue matrix instructions to tensor cores. Producer warp groups must efficiently load data to ensure that the consumer has sufficient data to keep the tensor cores fully utilized.

### 2.3 FLASHATTENTION FOR ACCELERATORS

Appendix section A discusses the basic attention computation. Efficient attention implementations (FlashAttention (Dao et al., 2022)) fuse the attention score computation and multiplication of attention scores with values for fast execution. Fig. 3a depicts how flash attention is implemented in a GPU. The kernel takes queries, keys, and values ($\mathbf{Q}, \mathbf{K}, \mathbf{V}$) matrices as input, and produces output matrix $\mathbf{O}$. The queries, keys, values, and output tokens form a sequence of $N$ tokens (gray bars), as shown in the figure. The figure shows a set of output tokens highlighted in green, computed by the threads of one block. This set of tokens is labeled $\mathbf{O}_{\text{tile}}$ highlighted in red. To compute $\mathbf{O}_{\text{tile}}$, the block first loads a corresponding set of queries $\mathbf{Q}_{\text{tile}}$ into shared memory ❶. Then, the first set of key and value tokens, $\mathbf{K}_{\text{tile}}$ ❷ and $\mathbf{V}_{\text{tile}}$ ❹, is loaded into shared memory. This tile is used to compute $\mathbf{Q}_{\text{tile}}\mathbf{K}_{\text{tile}}^T$ using the tensor core ❸, followed by exponentiation to compute $e^{\mathbf{Q}_{\text{tile}}\mathbf{K}_{\text{tile}}^T}$. This result stored in registers is then multiplied by the corresponding $\mathbf{V}_{\text{tile}}$ using the tensor core ❺. The result of the computation is added to $\mathbf{O}_{reg}$, a slice of output in registers ❻. In the next iteration, the next tile $\mathbf{K}_{\text{tile}}, \mathbf{V}_{\text{tile}}$ in the sequence is loaded into shared memory, and computation is repeated ❼. The sum over the exponents $e^{\mathbf{Q}_{\text{tile}}\mathbf{K}_{\text{tile}}^T}$ is also computed in registers to hold the denominator of the softmax function (not shown in figure).

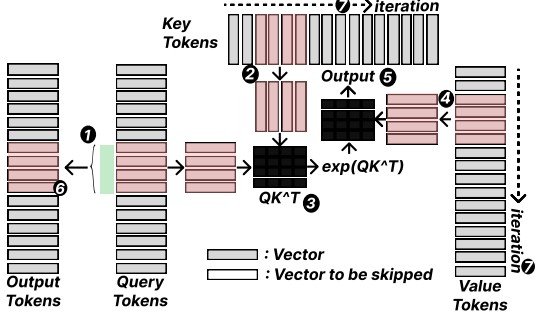
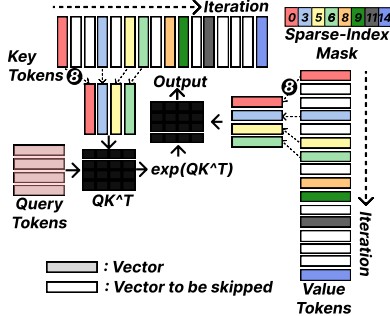

(a) FlashAttention: tiles of keys, values are loaded into shared memory. Tensor cores are used to compute a partial sum of the output vector.

(b) FG-Attn: only relevant keys and values corresponding to indices in sparse index mask are loaded into shared memory.

Figure 3: Implementation overview of FlashAttention (Dao et al., 2022) and FG-Attn.

## 3 ANALYSIS

### 3.1 VIDEO GENERATION TIMES

Video-DiT models encode video frames into latent embeddings using a vector-quantized VAE (VQ-VAE), which compresses each spatiotemporal patch of $H \times W$ pixels across $F$ frames into a single vector (e.g., $H = W = 8, F = 4$ maps every $8 \times 8$ block over 4 frames to one embedding). A five-second video at $480 \times 832$ resolution is thus represented by $\sim 32{,}000$ embeddings. Processing attention over

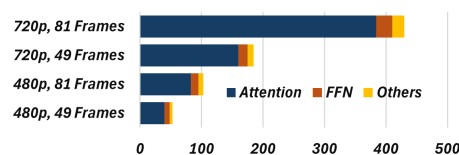

Figure 4: Seconds spent by different operations in Wan 2.1 1.3B (eager mode).

such long sequences requires massive computation, making attention the dominant cost even for short videos. Fig. 4 breaks down this runtime by operator ("others" includes VQVAE encoding of text tokens and initial noisy frames). The majority of time is spent in attention, which for a sequence of length $N$ scales as $O(N^2)$, unlike the feed-forward network ($O(N)$). As video resolution or length increases, attention dominates even more: in Wan 2.1 1.3B, attention accounts for 91% of runtime for 81 frames at 720p, compared to the already high 76% for 49 frames at 480p.

## 3.2 ATTENTION SPARSITY

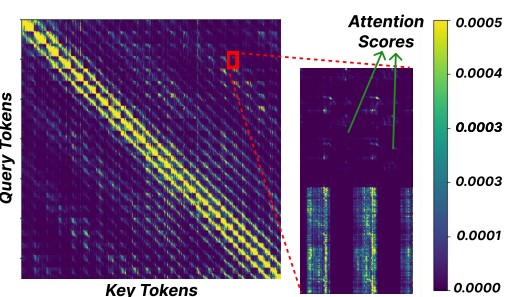

Figure 5: Sparsity in attention computation: attention scores are highly sparse and irregular.

Fig. 5 shows heatmap of the attention scores in one attention head of the Wan 2.1 1.3B vDIT model. We observe that the vast majority of attention scores are close to zero, over 85% in this example. Thus, attention computation can be significantly accelerated by bypassing these score computations. For sequence length $N$ and model dimension $D$, dense attention costs $O(N^2 D)$ floating-point operations (see section A). If only a fraction $\rho \in [0, 1]$ of the $N^2$ query–key pairs are retained (i.e., $\rho$ is the nonzero density, $\rho = 1 - \text{sparsity}$), the cost becomes $O(\rho N^2 D)$. Sparse attention mechanisms such as FlexAttention Dong et al. (2024), block-sparse attention Guo et al. (2024), and FlashAttention Dao et al. (2022) accelerate execution by skipping redundant attention score computations. They avoid loading and computing pairwise $q$–$k$ dot products over contiguous blocks of queries and keys (typically 64 each). The block size $M$ is fixed by GPU tensor-core dimensions (e.g., $64 \times 64$ on H100), so skipping one block avoids $M \times M$ dot products. To preserve accuracy, however, an entire block can be skipped only if *all* query–key pairs in it yield negligible scores.

**Sparsity in attention scores is fine-grained.** Table 1 reports attention map sparsity at different block sizes, measured as the fraction of $M \times M$ blocks with all scores below a threshold of $0.5/N$, where $N$ is the sequence length. Finer blocks ($16 \times 16$) yield about 70% sparsity, while coarser blocks ($64 \times 64$) achieve only 22%. This shows that finer granularity offers

Table 1: Sparsity vs. block size: % of $M \times M$ attention map blocks with all scores $\leq$ threshold.

| Block size | Sparsity | TFLOPs |
|---|---|---|
| $128 \times 128$ | 5.5% | 0.519 |
| $64 \times 64$ | 22.8% | 0.424 |
| $32 \times 32$ | 47.7% | 0.287 |
| $16 \times 16$ | 70.7% | 0.161 |

much greater opportunity for speedup, yet existing block-sparse implementations cannot exploit blocks smaller than $64 \times 64$. Our work targets this gap by exploiting finer-grained sparsity to design a more efficient attention mechanism that reduces FLOPs without loss of accuracy.

## 4 METHOD

In order to leverage fine-grained sparsity in DiTs, we must (1) implement a fine-grained sparse attention kernel on modern GPUs that skips computing slices attention scores as shown in Fig. 6, and (2) identify the slices of the attention map (mask) that can be skipped without sacrificing accuracy. To this end, we introduce FG-Attn, an efficient fine-grained sparse attention mechanism for DiTs. We now discuss its implementation.

### 4.1 REPRESENTING THE FINE-GRAIN SPARSE ATTENTION MASK

To implement FG-Attn, the attention mask must specify which slices of scores to compute, i.e., the key/value vectors required for each group of $M$ queries. For each group

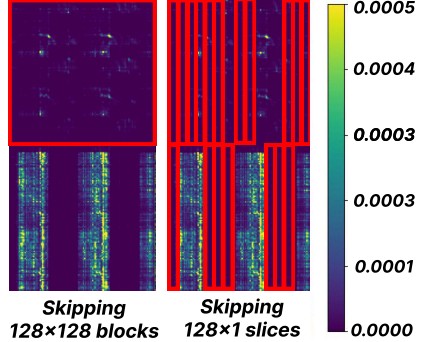

Figure 6: Block sparse attention mechanisms skip tiles of $128 \times 128$ attention map scores. We propose a method to skip fine-grain $128 \times 1$ sections.

of queries, we maintain an array of integer indices identi-
fying the keys/values to load from HBM for every attention
head. For $B$ batches, $H$ heads, $N/M$ query groups, and
up to $N$ keys, this mask is stored as a 4D integer array of shape $[B, H, N/M, N]$. The group size $M$
is chosen to match the tensor-core matrix multiplication unit, typically 64 or 128 on modern GPUs
(128 in our Hopper implementation).

### 4.2 FG-ATTN IMPLEMENTATION: GATHER-LOAD PRIMITIVE

In FlashAttention (Dao et al., 2022), a block of $M$ queries $\mathbf{Q}_{tile}$ and $M$ contiguous keys $\mathbf{K}_{tile}$ are
loaded into shared memory and multiplied using tensor cores. On GPUs, these loads are accelerated
by the Tensor Memory Accelerator (TMA). In FG-Attn, we instead must *gather* $M$ non-contiguous
keys (given by the sparse attention mask), pack them into a tile $\mathbf{K}_{rel}$, to compute $\mathbf{Q}_{tile}\mathbf{K}_{rel}^T$ (**8**
in Fig. 3b). To do this efficiently, we introduce a new *gather-load* primitive, which loads sparse
key/value vectors from HBM into shared memory based on the attention mask. The corresponding
value vectors are loaded with the same indices, and tensor cores compute partial sums across tiles
until all relevant slices are processed. Loading sparse key/value vectors requires first computing the
addresses of elements specified by the sparse index mask before issuing the load. Efficient gather-
load requires (i) fast address generation for sparse indices and (ii) hiding this latency. We achieve
(i) by parallelizing index-to-address translation across threads in a warp group and (ii) by pipelining
(i.e., overlapping) the gather-load with attention computation.

**Parallelizing address generation with the
sparse gather-load primitive.** The *gather-
load* primitive takes an array of indices,
fetches the corresponding key/value vectors
from HBM, and assembles them as a contigu-
ous tile in shared memory. For each group
of $M$ queries, only the relevant keys indi-
cated by the sparse mask are loaded (Fig. 7).
Indices are first cooperatively loaded into
shared memory **1** from HBM, then dis-
tributed across the four warps in a warp
group **2**. Each warp broadcasts its indices
to threads via warp-shuffle **3**, after which
threads compute addresses and issue asyn-
chronous loads to fetch the sparse vectors into shared memory **4**.

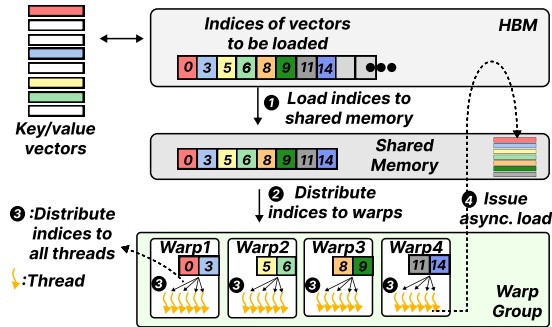

Figure 7: Detailed diagram showing the implemen-
tation of gather + load operation.

**Overlap address generation latency with at-
tention computation.** The latency of gen-
erating addresses from the sparse index mask
can be hidden behind attention computation. On
H100 GPUs, the *producer* warp groups load data
and *consumer* warp groups performs computa-
tion. Producers load query, key, and value tiles
into shared memory, while consumers compute
and accumulate partial sums (Fig. 8). At each it-
eration, the producer loads indices for $M$ queries

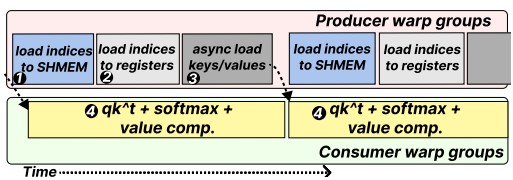

Figure 8: The address generation latency is
hidden by attention computation by having the
gath-load operation in the producer threads.

from the sparse mask into shared memory **1**, moves them into registers **2**, and issues asynchronous
loads for the corresponding rows from global memory **3**. Meanwhile, the consumer computes at-
tention scores and outputs **4**, fully hiding address-generation and load latency.

### 4.3 DETERMINING THE SPARSE-INDEX MASK

**Thresholding by caching attention mask across denoising iterations.** To exploit fine-grained
sparsity, we need a mask that identifies which query–key slices produce *significant* attention scores.
A straightforward approach would be to recompute this mask at every denoising iteration, but doing
so requires evaluating all attention scores, eliminating much of the benefit. Prior work (Hu et al.,
2025; Ma et al., 2025) has shown that intermediate embeddings change little across iterations, sug-
gesting that the sparsity pattern is stable. Inspired by this, we cache the sparse index mask obtained
in one iteration and reuse it in subsequent ones (Fig. 10). At timestep $t$, the mask for each head is

taken from scores exceeding a threshold in the previous step $(t-1)$. In practice, we compute the mask once by evaluating full scores and marking slices with at least one score above $\tau_{cached}$, then store it in HBM per head for reuse.

Fig. 9 confirms mask stability for Wan 2.1 (Wan et al., 2025) 1.3B and 14B. We compare masks across 5 denoising steps and measure the flip rate, i.e., the fraction of positions whose scores cross $\tau$ between steps. For example, the 5–15 label shows the percentage of entries $\leq \tau$ at step 5 but $> \tau$ at step 15. Over 96% of scores remain below $\tau$ once they are below in the previous iteration, demonstrating that the cached sparse mask is highly stable and can be reused efficiently.

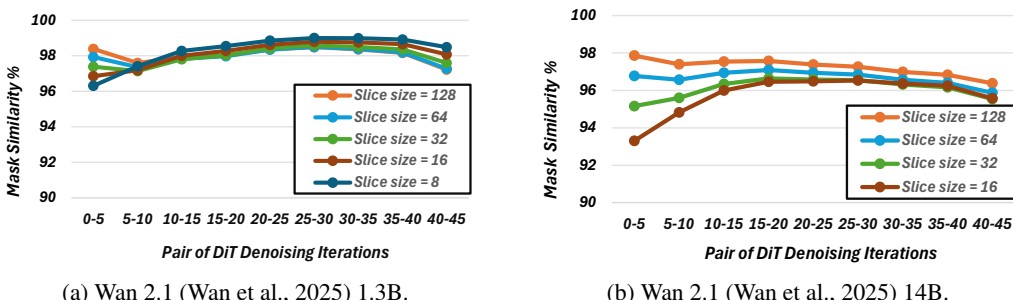

(a) Wan 2.1 (Wan et al., 2025) 1.3B.  (b) Wan 2.1 (Wan et al., 2025) 14B.

Figure 9: % of attention scores that newly cross the threshold (i.e., differences in the sparse index mask) between 5 denoising iterations. Shown for slice sizes $128 \times 1$, $64 \times 1$, $32 \times 1$, $16 \times 1$, and $8 \times 1$.

**Thresholding based on average-query.** In the context of video diffusion models, $q_1, q_2, \ldots, q_M$ are query tokens corresponding to adjacent pixels in space and time. Such adjacent tokens typically exhibit similar responses compared to their surrounding queries. Motivated by this observation, we propose a simple, lightweight strategy for determining the attention mask. We compute the average of a group of queries, $q_{avg} = (q_1 + q_2 + .. + q_M)/M$. A key $k$ is included if its dot product with $q_{avg}$ is significant. We then apply a threshold over these averaged scores. Specifically, the threshold $\tau$ is computed as $\exp(k.q_{avg}/\sqrt{D})/D < \tau_1$.

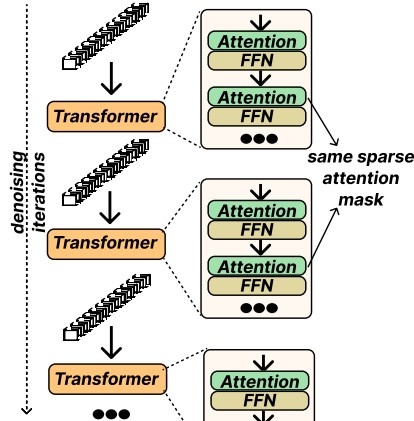

Figure 10: Determining attention mask based on attention scores observed in the previous denoising iteration.

## 5 RESULTS

### 5.1 METHODOLOGY

We evaluate FG-Attn using the following open source, widely available video models: (1) Wan 2.1 (Wan et al., 2025) 1.3B, 14B models at 480p and 720p, at 81 frames, and (2) HunyuanVideo (Kong et al., 2024) at 720p. 81 frames. All experiments are conducted using bfloat16 precision. We implement two versions of FG-Attn: (1) by using device primitives from ThunderKittens (Spector et al., 2024) and (2) integrated into FlashAttention-3 (Shah et al., 2024) for an H100 GPU. To evaluate the quality of the videos generated, we evaluate the quality of the videos generated using peak signal-to-noise ratio (PSNR) and structural similarity (SSIM), and measure their absolute video generation times. We also evaluate the videos we generated using the VBench (Huang et al., 2024) VLM benchmarking scores, alongside visual comparisons of frames from the generated videos in the Appendix section C.1. We test two configurations of FG-Attn: one using the caching strategy to determine the mask (FG-Attn-cached), and the other using the pooling strategy (FG-Attn-pooling). For the FG-Attn-cached strategy, the threshold is set to $0.5/N$, where $N$ is the number of embedding vectors in the latent space representation of the video. The attention mask is cached once every 15 DiT iterations. We compare FG-Attn with two prior works that use block sparse attention to leverage sparsity in attention scores in DiTs: Radial Attention (Li et al., 2025), SparseVideo-Gen (Xi et al., 2025), SparseVideo-Gen2 (Yang et al., 2025), and SpargeAttn (Zhang et al., 2025b).

### 5.2 END-TO-END SPEEDUP

Fig. 11a shows the end-to-end speedup on video generation times, normalized to baseline. We observe that FG-Attn is able to achieve an average speedup of $1.48\times$ and up to $1.65\times$. FG-

Attn achieves a speedup as a result of accelerating the attention computation time during training. Fig. 11b shows the average runtime needed to compute the attention of every layer, normalized to the PyTorch implementation baseline. For the attention computation, FG-Attn achieves a speedup of $1.93\times$ on average, up to $2.38\times$. FG-Attn achieves a higher speedup when generating videos at 720p. Our approach achieves a higher speedup of $1.2\times$ compared to SparseVideoGen (Xi et al., 2025) and $1.22\times$ compared to RadialAttention (Li et al., 2025). The observed speedup comes from skipping a larger fraction of attention scores. However, this advantage diminishes at higher video resolutions (720p compared to 480p). This is because, in self-attention, interactions between blocks of embeddings that correspond to distant regions of the video are typically zero. As the resolution increases, each embedding vector covers a smaller region of the input, leading to a greater number of embeddings. This increases the proportion of zero-valued attention scores, which block-sparse attention can skip. Consequently, while more scores are skipped, the relative speedup achieved by FG-Attn decreases.

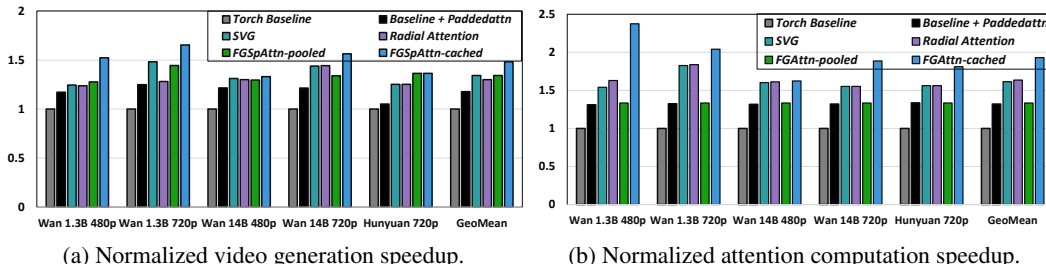

(a) Normalized video generation speedup.  (b) Normalized attention computation speedup.

Figure 11: End-to-End and attention computation speedup results for video generation models.

## 5.3 Absolute Video Generation Times vs. Quality

In this section, we compare absolute wall clock time and video quality across various prior works. We generated videos for prompts from the Penguin Benchmark (Penguin Benchmark, 2024) and generate videos corresponding to the first 21 prompts in the benchmark for every video model. For a fair comparison, among all training-free acceleration methods, we set the number of "warmup" steps (initial steps running full attention) to be equal (24% of inference steps), similar to prior work (Yang et al., 2025). We note that for some open-source prior works, the baseline Flash Attention (FA) is different for warmup versus inference (e.g., in SVG2). Thus, we specify for each configuration which FA version is used for the two phases. We list the experiment configurations as follows:

- **FA2-based Benchmarks:** We compare against dense FA2 (Dense) (Dao, 2023), Radial Attention (Li et al., 2025), and SparseVideoGen (SVG) (Xi et al., 2025), all of which utilize FA2 for both warmup and sparse operations. We also include the open-source SparseVideoGen2 (SVG2) (Yang et al., 2025), which uses FA2 for warmup and FlashInfer's FA3 (Ye et al., 2025) attention kernel for sparse attention. Ours in this setting utilizes FA2 for warmup with the attention kernel implemented via ThunderKittens implementation of FA3 (Spector et al., 2024) (TK-FA3).

- **FA3-based Benchmarks:** We compare against dense FA3 (Shah et al., 2024), SpargeAttention (Zhang et al., 2025b)(FA3 warmup with FP8 attention), and a FlashInfer's FA3-based implementation of SVG2 (Yang et al., 2025) (utilizing FlashInfer). Ours is implemented entirely over FA3. We also test a hybrid variant, Ours + SVG2, which applies SVG2 clustering permutation to queries before executing our FA3-based attention.

Tables 2, 3, 4 list the wall-clock time and PSNR/SSIM quality of the videos generated for the Wan-14B, Wan 1.3B and HunyuanVideo video model respectively. We make the following observations. First, on Wan-14B 720p, Wan 1.3B (720p/480p) and HunyuanVideo, while FG-Attn is slower, it outperforms SVG and Radial Attention in terms of video quality in the produced output video. The slowdown is because the ThunderKittens implementation of FA3 incurs higher overhead for longer sequences. The FA3 implementation of FG-Attn outperforms both the mechanisms. Second, SpargeAttn can generate videos much faster for every video model because it computes attention at lower precision (SpargeAttn for H100 uses 8-bit quantization, whereas SVG2/ours uses bf16) which compromises quality. Quantization is orthogonal to both FG-Attn and SVG2, and can be incorporated into these implementations as well. For HunyuanVideo, SpargeAttn produces significantly different videos when compared to the baseline, leading to significantly lower PSNR/SSIM. Third, FG-Attn achieves higher speedups compared to SVG2 (at 16-bit precision) on Wan 14B and Wan 1.3B at 480p resolution. While Sparse-VideoGen 2 outperforms FG-Attn at 720p resolution

on HunyuanVideo, Wan 14B, and Wan 1.3B, it does so by only a small margin, despite a much larger reduction in FLOPs. This is because SVG2 incurs significant overhead from computing query and key clusters, which becomes a larger portion of the total runtime on faster models. However, reordering query tokens into clusters before applying FG-Attn (denoted as Ours+SVG2) results in performance that exceeds SVG2 across all baselines.

Table 2: Wall clock time and quality results on the Wan 14B model Wan et al. (2025)

| | Wan 14B 720p | | | | Wan 14B 480p | | | |
|---|---|---|---|---|---|---|---|---|
| | PSNR | SSIM | Time (mins) | FLOPs | PSNR | SSIM | Time (mins) | FLOPs |
| Dense (FA2) | inf. | 1.0 | ∼28 | 1X | inf. | 1.0 | 7:30 | 1X |
| Radial (FA2+FA2) | 21.1 | 0.72 | 20:45 | 0.48X | 22.01 | 0.76 | 6:24 | 0.48X |
| SVG (FA2+FA2) | 19.82 | 0.70 | 17:32 | 0.47X | 20.13 | 0.74 | 5:18 | 0.47X |
| Ours (FA2+TK-FA3) | 23.01 | 0.79 | 18:29 | 0.68X | 22.04 | 0.78 | 5:41 | 0.68X |
| Dense (FA3) | inf. | 1.0 | 16:45 | 1X | inf. | 1.0 | 4:12 | 1X |
| Sparge (FA3+fp8) | 22.51 | 0.79 | 13:04 | - | 20.11 | 0.78 | 3:58 | - |
| SVG2 (FA3+FA3) | 22.39 | 0.78 | 15:25 | 0.47X | 23.48 | 0.84 | 4:58 | 0.47X |
| Ours (FA3+FA3) | 23.01 | 0.79 | 15:55 | 0.68X | 22.04 | 0.78 | 4:02 | 0.68X |
| Ours+SVG2 (FA3) | 22.93 | 0.78 | 15:14 | 0.53X | 22.31 | 0.78 | 4:01 | 0.53X |

### 5.4 ABLATION STUDY

**Latency overheads of FG-Attn's sparse attention kernel.** Fig. 12 shows the runtime of FG-Attn's sparse attention kernel under varying sparsity levels, normalized to baseline dense attention. First, at 0% sparsity, runtime of FG-Attn is within 5% of the dense attention implemented in ThunderKittens (Spector et al., 2024). Since FG-Attn is built on top of ThunderKittens' dense kernel, its gather-load address generation is effectively hidden behind the attention computation. Second, the runtime decreases linearly with sparsity. Thus FG-

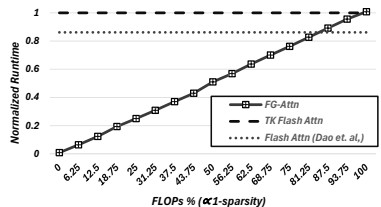

Figure 12: Normalized runtime of FG-Attn's sparse attention kernel vs sparsity.

Attn's time complexity scales with FLOPs as $O(N^2D(1 - \text{sparsity}))$, where $N$ is the sequence length and $D$ is the model dimension (see section 3.2). Third, both ThunderKittens dense attention and FG-Attn 's sparse kernel are 14% slower than FlashAttention as FlashAttention uses sophisticated optimizations (e.g., ping-pong scheduling of softmax and tensor-core ops) to increase utilization. Since we leave tensor-core scheduling unchanged, these optimizations can be adopted in FG-Attn and narrow the performance gap.

**Attention computation times on varying threshold.** Fig. 13 depicts the average attention computation time for video generation as the threshold parameter is varied. We sweep the threshold parameter from $0.1/N$ to $1/N$, where $N$ is the number of embedding vectors in the latent space representation of the video. We observe that across all video models, the attention computation A higher threshold enables skipping a larger amount of computation, thereby leading to a speedup.

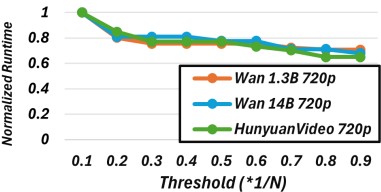

Figure 13: Normalized attention computation time at different thresholds.

### 5.5 BROADER APPLICABILITY

Although we evaluate FG-Attn primarily on video diffusion, the same techniques apply to any long-context diffusion model. To illustrate, we apply FG-Attn to DiTs for $2048 \times 2048$ image generation. Table 5 reports end-to-end runtimes and attention speedups, showing that FG-Attn accelerates the Flux model by $1.23\times$ over torch baseline.

Table 5: Flux DiT: Attention and Generation Speedup.

| | Attn | Gen. |
|---|---|---|
| Baseline | 1X | 1X |
| Padded Attn | 1.4X | 1.1X |
| FG-Attn-cache | 1.77X | 1.23X |

## 6 RELATED WORK

**Block-sparse attention.** Several implementations of block-sparse attention (Guo et al., 2024; Dao et al., 2022; Dong et al., 2024; Ye et al., 2025; Wang et al., 2024) propose coarse-grained mech-

Table 3: Wall clock time and quality results on the Wan 1.3B model Wan et al. (2025)

| | Wan 1.3B 720p | | | | Wan 1.3B 480p | | | |
|---|---|---|---|---|---|---|---|---|
| | PSNR | SSIM | Time (mins) | FLOPs | PSNR | SSIM | Time (mins) | FLOPs |
| Dense (FA2) | inf. | 1.0 | 06:46 | 1X | inf. | 1.0 | 1:38 | 1X |
| SVG (FA2+FA2) | 22.40 | 0.78 | 4:01 | 0.47X | 16.70 | 0.64 | 1:07 | 0.47X |
| Ours (FA2+TKFA3) | 27.03 | 0.85 | 4:09 | 0.59X | 21.55 | 0.77 | 1:08 | 0.605X |
| Dense (FA3) | inf. | 1.0 | 2:53 | 1X | inf. | 1.0 | 1:01 | 1X |
| Sparge (FA3+f8) | 27.10 | 0.84 | 2:26 | - | 21.41 | 0.79 | 00:43 | - |
| SVG2 (FA3+FA3) | 27.60 | 0.87 | 2:30 | 0.39X | 22.44 | 0.82 | 1:00 | 0.39X |
| Ours (FA3+FA3) | 27.03 | 0.85 | 2:14 | 0.59X | 21.55 | 0.77 | 00:51 | 0.605X |
| Ours+SVG2 (FA3+FA3) | 27.03 | 0.87 | 2:10 | 0.45X | 21.61 | 0.77 | 00:58 | 0.44X |

Table 4: Wall clock time and quality results on the HunyuanVideo model Kong et al. (2024)

| | HunyuanVideo 720p | | | |
|---|---|---|---|---|
| | PSNR | SSIM | Time (mins) | FLOPs |
| Dense (FA2) | inf. | 1.0 | 12:38 | 1X |
| Radial (FA2) | 22.33 | 0.77 | 09:07 | 0.48 X |
| SVG (FA2) | 20.19 | 0.76 | 06:01 | 0.45X |
| Ours (FA2+TK-FA3) | 24.02 | 0.80 | 07:32 | 0.68X |
| Dense (FA3) | inf. | 1.0 | 06:56 | 1X |
| Sparge (FA3+fp8) | - | - | 05:04 | - |
| SVG2 (FA3+FA3) | 25.80 | 0.87 | 06:24 | 0.44X |
| Ours (FA3+FA3) | 24.02 | 0.80 | 06:30 | 0.68X |
| Ours+SVG2 (FA3) | 24.8 | 0.84 | 06:11 | 0.54X |

anisms that skip entire tiles of attention scores, typically at $64 \times 64$ or $128 \times 128$ granularity in half-precision. These methods have been widely adopted in LLM inference (Jiang et al., 2024; Xu et al., 2025; Hong et al., 2023; Yuan et al., 2025; Gao et al., 2024). However, current block-sparse methods cannot operate at smaller tile sizes: reducing block size either fails to compile or leads to severe hardware underutilization due to tensor-core width constraints (section 3.2). For video diffusion, Radial Attention (Li et al., 2025), X-Attention (Xu et al., 2025), STA (Zhang et al., 2025d), SparseVideoGen (Xi et al., 2025), and SparseVideoGen2 (Yang et al., 2025) employ fixed sparsity patterns based on empirical observations. SpargeAttention (Zhang et al., 2025b) predicts a mask based on pooling blocks of queries and keys. Video Sparse Attention (VSA) (Zhang et al., 2025e) and VMoBA (Wu et al., 2025) learn to predict the mask. All these methods, however, are limited to coarse-grained block skipping. We directly compare against SparseVideoGen and Radial Attention in section 5. Furthermore, trainable approaches such as VSA (Zhang et al., 2025e) can be reformulated to generate masks compatible with FG-Attn, and are thus orthogonal and complementary.

**Other techniques to accelerate video diffusion.** SageAttention (Zhang et al., 2024b;a; 2025c;a), use quantization uses quantization to speedup attention layers in transformers. Since quantization and sparsity are orthogonal, FG-Attn can be applied on top of quantization-based approaches. Caching-based approaches such as DeepCache Ma et al. (2024), TeaCache Liu et al. (2024) and TaoCache (Fan et al., 2025) exploit temporal redundancy across denoising steps. These approaches are orthogonal to our work and can be combined with FG-Attn for additional acceleration.

# 7 CONCLUSION

We introduced **FG-Attn**, a fine-grained sparse attention mechanism that skips redundant query–key computations at slice-level granularity. By combining a hardware-efficient gather-load primitive with lightweight, training-free mask generation strategies, FG-Attn enables practical exploitation of fine-grained sparsity on modern GPUs. Applied to state-of-the-art video and image diffusion transformers, FG-Attn achieves significant end-to-end acceleration with negligible quality loss. These results demonstrate that fine-grained sparsity can be realized efficiently, providing a scalable alternative that subsumes block-sparse attention in DiTs.

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

## A    ATTENTION COMPUTATION

For a set of $N$ queries $\mathbf{q_1}, \mathbf{q_2}, ...\mathbf{q_N}$, $N$ keys $\mathbf{k_1}, \mathbf{k_2}, ...\mathbf{k_N}$ and $N$ values $\mathbf{v_1}, \mathbf{v_2}, ...\mathbf{v_N}$ of dimension $D$, the self-attention layer computes, at each attention head,

$$\mathbf{O} = \text{softmax}\left(\frac{\mathbf{QK}^T}{\sqrt{D}}\right)\mathbf{V} \tag{1}$$

Where $\mathbf{Q}, \mathbf{K}, \mathbf{V}$ are $N \times D$ matrices consisting of query, key, and value vectors, respectively. $\mathbf{O}$ is the output matrix of size $N \times D$. The time-complexity of computing attention grows quadratically with the sequence size $N$, as $O(N^2 D)$, resulting from computing a 2D matrix $\mathbf{QK}^T$ matrix of size $N \times N$. This 2D matrix, computed from the query, key matrices ($\mathbf{Q}, \mathbf{K}$) followed by a softmax operation is referred to as the *attention map*. Each element of the attention map is called the *attention score*. The expression for the attention map and attention scores (indexed by i, j) is given by:

$$\text{softmax}\left(\frac{\mathbf{QK}^T}{\sqrt{D}}\right) \qquad a_{ij} = \frac{e^{\mathbf{q}_i \mathbf{k}_j}}{\sum_{n=1}^{N} e^{\mathbf{q}_i \mathbf{k}_n}}$$

The memory footprint when computing attention naively is $O(N^2)$, where $N$ is the sequence length. This is the result of computing and storing $\mathbf{QK}^T$ (Eq. 1), which requires materializing an $N \times N$ matrix for each attention head in memory. This becomes problematic when computing attention for long sequences, with the size of this intermediary $\mathbf{QK}^T$ matrix often exceeds the accelerator's HBM capacity. Efficient implementations of attention in GPUs (flash attention Dao et al. (2022)) avoids this high memory footprint by fusing the attention score computation and the multiplication of attention map with the value matrix.

## B    GPU ARCHITECTURE OVERVIEW

Fig. 14 provides a high-level overview of modern GPU architecture. Tensor data is first transferred from high-bandwidth memory (HBM) into the shared memory of streaming multiprocessors (SMs), from which threads schedule computations on the tensor cores

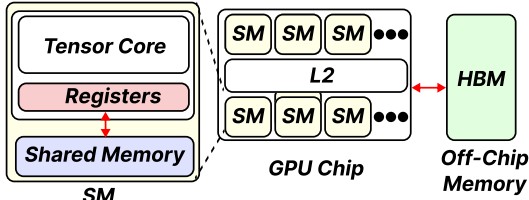

Figure 14: High-level view of modern GPU architecture.

Threads within a GPU thread block are divided into producers and consumers, as shown in Fig. 15. Producer threads issue load operations to move data from HBM into shared memory, while consumer threads schedule computations on the tensor cores using the fetched data. By overlapping these operations, GPU resources remain efficiently utilized. This pipelined execution is illustrated in Fig. 15.

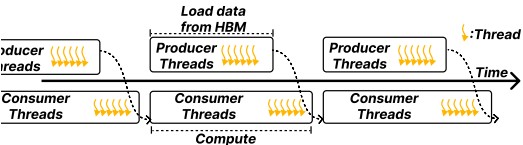

Figure 15: Pipelined execution of producer and consumer threads: Data is prefetched by the producer while the consumer threads are doing computation.

## C  VIDEO DIFFUSION MODELS

Figs. 16, 17 and 18 show the visual representation of the produced video compared to the original (the top row of each set of videos represents the baseline video) for the HunyuanVideo model, Wan 1.3B model, and the Wan 14B model, respectively.

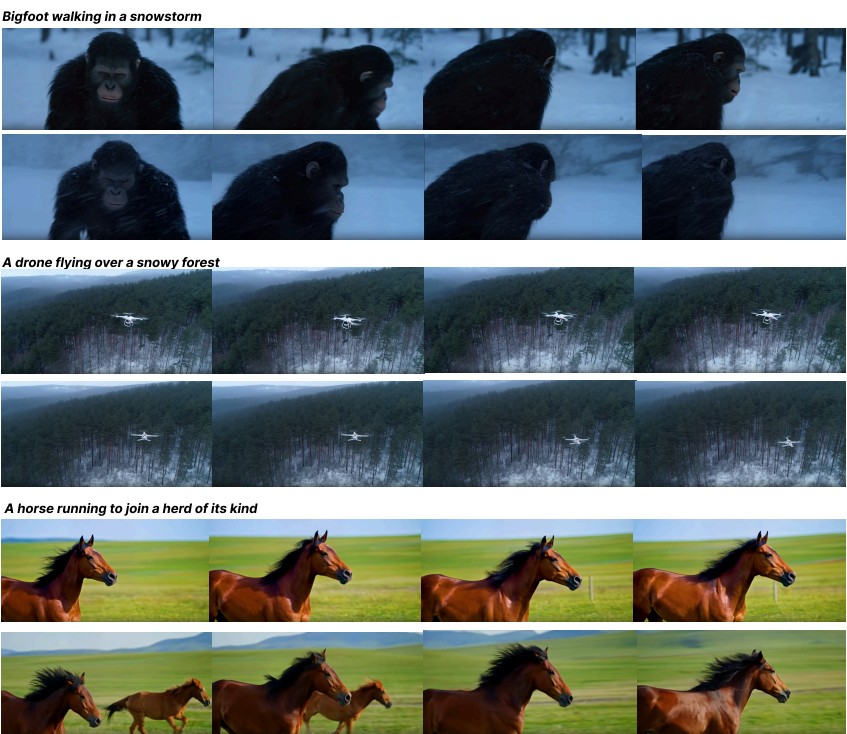

Figure 16: Samples of videos generated using baseline HunyuanVideo model, and FG-Attn-HunyuanVideo (The baseline generates first row, second row generated using FG-Attn)

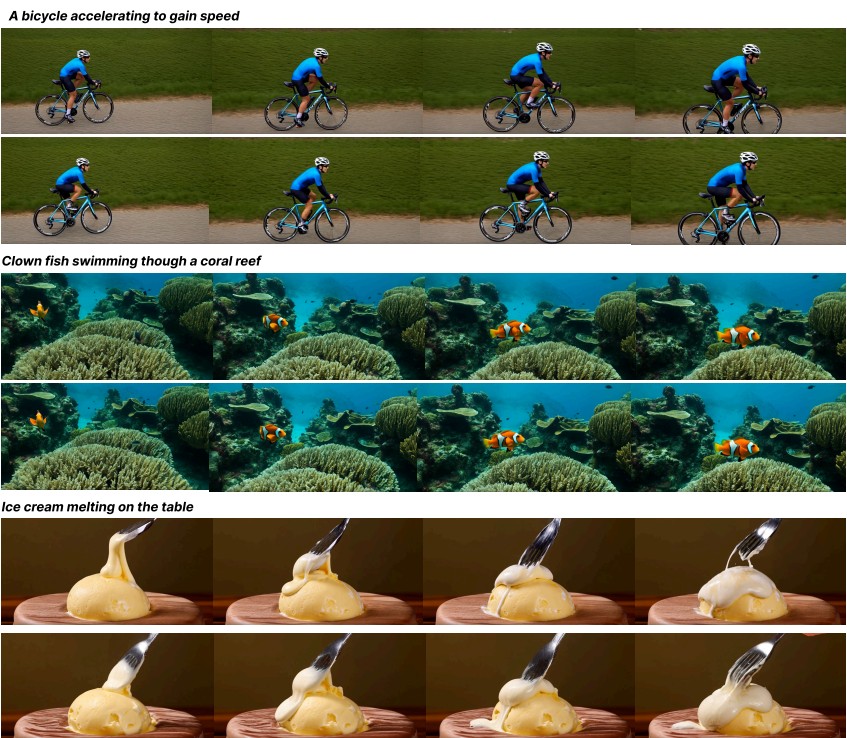

Figure 17: Samples of videos generated using baseline Wan-1.3B model, and FG-Attn-Wan1.3B. (First row is generated by the baseline, second row is generated using FG-Attn)

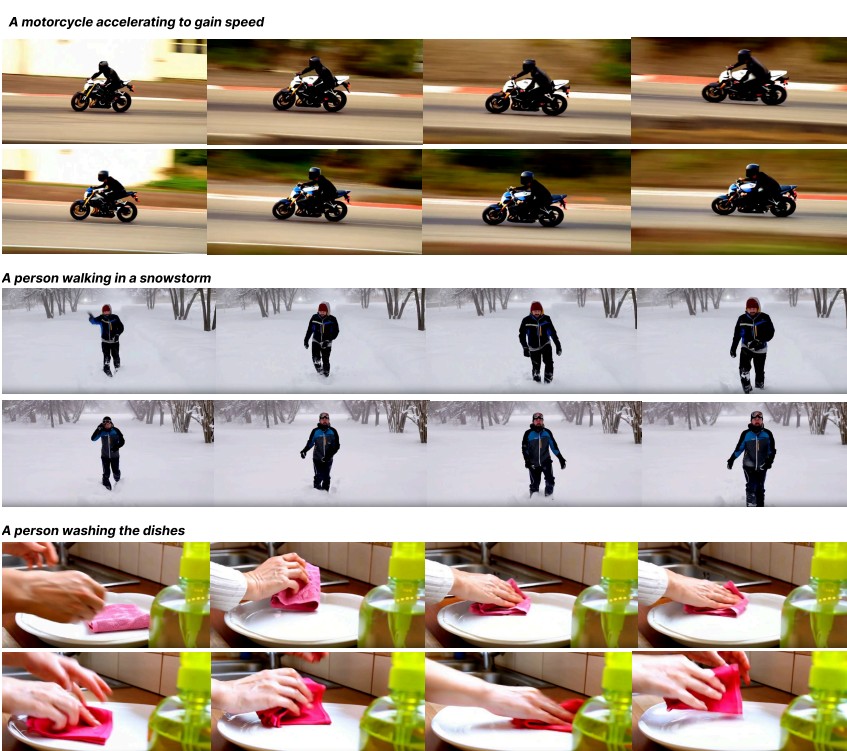

Figure 18: Samples of videos generated using baseline Wan-14B model, and FG-Attn-Wan14B (First row is generated by the baseline, second row is generated by FG-Attn)

## C.1 QUALITATIVE ANALYSIS USING VBENCH

Table 6 shows the VBench (Huang et al., 2024) video benchmarking results when compared to the baseline. On VBench, FG-Attn achieves negligible degradation in quality. The Appendix section C show the visual representation of the produced video compared to the original (the top row of each set of videos represents the baseline video) for the HunyuanVideo model, Wan 1.3B model, and the Wan 14B model, respectively. We find that across all the prompts tested here, FG-Attn can recover the original video with no quality degradation. FG-Attn also retains the generated video style and does not significantly shift the distribution captured by the underlying model.

Table 6: Visual quality of the generated video evaluated using VBench Huang et al. (2024)

|  | *Aesthetic Quality* | *Subject Consistency* | *Background Consistency* | *Overall Consistency* |
|---|---|---|---|---|
| Wan-1.3B 480p baseline | 0.601 | 0.936 | 0.958 | 0.23 |
| Wan-1.3B 480p FGAttn | 0.605 | 0.939 | 0.96 | 0.23 |
| Wan-1.3B 720p Baseline | 0.61 | 0.944 | 0.962 | 0.233 |
| Wan-1.3B 720p FGAttn | 0.61 | 0.944 | 0.964 | 0.232 |
| Wan-14B 480p baseline | 0.623 | 0.953 | 0.97 | 0.25 |
| Wan-14B 480p FGAttn | 0.616 | 0.952 | 0.975 | 0.247 |
| Wan-14B 720p baseline | 0.621 | 0.945 | 0.969 | 0.248 |
| Wan-14B 720p FGAttn | 0.619 | 0.942 | 0.961 | 0.245 |
| Hunyuan-13B 720p baseline | 0.62 | 0.944 | 0.962 | 0.239 |
| Hunyuan-13B 720p FGAttn | 0.62 | 0.94 | 0.962 | 0.239 |

# D    USE OF LLMS

In this work, LLMs were used to polish sections of the writing and to check grammar in the draft. They also provided partial assistance in code development through extensions similar to GitHub Copilot.

