# OpenReview forum: "FG-ATTN: LEVERAGING FINE-GRAINED SPARSITY IN DIFFUSION TRANSFORMERS"
_ICLR.cc/2026/Conference — ICLR 2026 Conference Desk Rejected Submission_

### Official Review · Reviewer_Esgn · 2025-10-28

**Soundness:** 3
**Presentation:** 2
**Contribution:** 2
**Rating:** 2
**Confidence:** 4

**Summary:**

The paper proposes FG-ATTN, a fine-grained attention mechanism for efficient video diffusion transformers. Instead of using coarse block-sparse attention, the method divides attention into smaller units and reorganizes tokens to better exploit computation efficiency. It also introduces a mean-query aggregation step to reduce redundant attention computation among neighboring tokens. Experiments on large video models show some speedup at the attention operator level. The paper claims that the method achieves around 1.5× acceleration without retraining.

**Strengths:**

- The topic is relevant and important for scaling video transformers efficiently.
- The paper provides a clean implementation idea for fine-grained attention.
- The writing is overall understandable, and the motivation for improving attention sparsity is reasonable.

**Weaknesses:**

**1.The paper mainly introduces two technical ideas, both of which are reasonable but lack some novelty.**

  - The fine-grained attention implementation can be simplified. Using a token permutation method for the key matrix is enough to achieve the same result, so the proposed gather-load process is unnecessarily complicated. This fine-grained attention idea is also not new. SVG2[1] already proposed a more fine-grained sparse attention than FG-Attn using token reordering to better utilize Tensor Core efficiency. SparseAttn[2] also applied token reordering to improve sparsity in block-sparse masks.

  -  The observation that adjacent tokens usually have similar responses and thus can share a mean query (qmean) is reasonable but not new. SpargeAttn[2] already reported this observation, and several other methods, such as MInference[3] and SeerAttention[4], have used mean pooling combined with thresholding to predict sparse attention masks. Additionally, pooling all blocks of Q and then multiplying them by all tokens in K will incur significant overhead.

**2. The experiments are incomplete and not convincing.**

  - On effectiveness, the end-to-end video quality should be compared with some baselines, but there is *no one baseline compared*.

  - On efficiency, the paper compares speedup results without showing the corresponding video quality. Without evaluating end-to-end acceleration while maintaining similar quality (or at least reporting the quality of each method), the claimed improvement is not meaningful. Reporting speedup alone is misleading, since higher speed can easily come at the cost of lower generation quality.

3. The work does not compare against strong and relevant baselines such as SVG2[1] or SparseAttn[2], both of which already achieve high attention speedup and maintain quality.

4. The abstract should be within one paragraph.


[1] Sparse VideoGen2: Accelerate Video Generation with Sparse Attention via Semantic-Aware Permutation

[2] SpargeAttention: Accurate and Training-free Sparse Attention Accelerating Any Model Inference

[3] MInference 1.0: Accelerating Pre-filling for Long-Context LLMs via Dynamic Sparse Attention

[4] SeerAttention: Learning Intrinsic Sparse Attention in Your LLMs

**Questions:**

Please refer to the weaknesses section for detailed questions and suggestions.

---

> ### Author Response · Authors · 2025-11-28
> **Response to review (1/2)**
>
> Dear reviewer,
>
> Thank you for your comments and feedback\! Please find the answers to your questions below.
>
> **Novelty of two technical ideas:**
>
> - Token permutation (introduced in SVG2): While token reordering promotes sparsity, as we explain, it does not fully leverage the sparsity that FG-Attn does. In SVG2, similar groups of keys and queries are clustered together to promote sparsity by grouping zero attention scores together. However, this results in fine-grained key clusters of varying sizes. Processing these irregular clusters prevents optimal Tensor Core utilization. FG-Attn avoids this limitation by requiring only query token reordering, leaving key and value tokens in their original order. We incorporate token reordering (similar to SVG2) into FG-Attn (denoted as FG-Attn+SVG2), where we only reorder the query tokens before computing attention. This approach outperforms both SVG2 and FG-Attn. To the best of our knowledge, SpargeAttn does not perform token reordering.
> - **Novelty of the pooling mechanism:** Prior mechanisms such as SpargeAttn, MInference, and SeerAttention compute attention scores over mean-pooled blocks of queries and keys to determine the block-sparse attention mask. However, MInference and SeerAttention show that this technique works primarily for applying block-sparse attention in Large Language Models (LLMs). Applying the same strategy to video generation models is insufficient; block-sparse attention scores computed over mean-pooled queries and keys alone fail to determine critical attention scores without requiring additional fine-tuning (VSA \[H\]). SpargeAttn applies a threshold on semantic similarity within the mean-pooled query and key blocks to determine the mask more accurately. FG-Attn allows us to perform this mechanism without relying on a threshold on the semantic similarity of keys. Thus, we make the novel observation that mean-pooling queries alone is sufficient to generate coherent videos using DiTs.
> - **Overhead of the pooling mechanism.** Thresholding attention scores computed over mean-pooled queries and keys does not significantly affect performance, as the number of operations required to compute attention scores is M times lower (where M is the size of the slice). Standard attention computation requires O(N^2D) operations, where N is the sequence length and D is the model dimension. In contrast, computing attention scores over pooled queries requires only O(N^2D/M) FLOPs. As M=128 in our experiments, this FLOP count is significantly smaller compared to the full attention computation.
>
> **Video quality comparison to baselines:** We have added results for  the quality of the videos generated and with their absolute video generation times. We now report PSNR and SSIM metrics alongside wall-clock inference time for all methods in the additional results below.
>
> **Comparison to SVG2 and SpargeAttn:** We note that the code for SparseVideoGen2 work was not available prior to the conference submission deadline. Additionally, SVG2 was not a published work at the time of submission (even the arxiv version was only just available). Thus, Sparse-Videogen2 should be considered a concurrent work by ICLR policies. Even so, we provide a detailed quantitative and qualitative comparison. We also compare against SpargeAttn, but we note that this is not an apples-to-apples comparison, since it uses fp8 precision while SVG2 and FG-Attn use bf16).
>
> We compare FG-Attn with SVG2 and SpargeAttn below.  *We also note that the optimizations in SVG2 are orthogonal to FG-Attn*. Thus, we have implemented a new configuration, “Ours+SVG2” that combines the two methods. Specifically, SVG2 permutes similar query and key tokens into clusters. For “Ours+SVG2”, we permute the query tokens such that clusters of similar queries are close by. This promotes sparsity in 128x1 queries/keys slices. This is because all the elements of the slices have either high or insignificant attention scores with respect to a key, thus reducing redundant computation.
>
> We consider the following configurations in our experiments:
>
> - Dense (FA3): Using FA3 \[E\].
> - SpargeAttention (FA3+fp8): Warmup implemented using FA3+Sparge Attention uses fp8 attention
> - SparseVideoGen2 (FA3+FA3): Warmup implemented using FA3+SVG2 on top of FA3 \[E\] and the SVG2 kernel implemented on FlashInfer’s FA3 \[B\].
> - Ours (FA3 \+ FA3): Our attention mechanism with the warmup steps implemented and sparse attention kernel implemented over FA3 \[E\]
> - Ours \+ SVG2 (FA3+FA3): FG-Attn mechanism built on top of FA3\[E\]. Before each attention run, the queries are permuted according to SVG2 clustering strategy.
>
> (continued)

---

> > ### Author Response · Authors · 2025-11-28
> > **Response to review (2/2)**
> >
> > |  | Wan 14B 720p |  |  |  | Wan 14B 480p |  |  |  |
> > | :---- | ----- | :---- | :---- | :---- | ----- | :---- | :---- | :---- |
> > |  | **PSNR** | **SSIM** | **Generation Time** | **Attn TFLOPs** | **PSNR** | **SSIM** | **Generation Time** | **Attn TFLOPs** |
> > | Dense (FA3) | inf. | 1.0 | 16:45 mins | 1X | inf. | 1.0 | 4:12 mins | 1X |
> > | SpargeAttention (FA3 warmup, f8 inference) | 22.51 | 0.79 | 13:04 mins | \- | 20.11 | 0.78 | 3:58 mins | \- |
> > | SparseVideoGen2 (FA3+ FA3) | 22.64 | 0.75 | 15:25 mins | 0.47X | 23.48 | 0.84 | 4:58 mins | 0.47X |
> > | *Ours (FA3+FA3)* | 23.01 | 0.79 | 15:55 mins | 0.68X | 22.04 | 0.78 | 4:02 mins | 0.68X |
> > | *Ours \+ SVG2 (FA3)* | 22.93 | 0.78 | 15:14 mins | 0.53X | 22.31 | 0.78 | 4:01 mins | 0.53X |
> >
> > |  | Wan 1.3B 720p |  |  |  | Wan 1.3B 480p |  |  |  |
> > | :---- | ----- | :---- | :---- | :---- | ----- | :---- | :---- | :---- |
> > |  | **PSNR** | **SSIM** | **Generation Time** | **TFLOPs** | **PSNR** | **SSIM** | **Generation Time** | **TFLOPs** |
> > | Dense (FA3) | inf. | 1.0 | 2:53 mins | 1X | inf. | 1.0 | 1:01 mins | 1X |
> > | SpargeAttention (FA3 warmup, f8 inference) | 27.10 | 0.84 | 2:26 mins | \- | 21.41 | 0.79 | 00:43 mins | \- |
> > | SparseVideoGen2 (FA3 \+ FA3) | 27.60 | 0.87 | 2:30 mins | 0.39X | 22.44 | 0.82 | 1:00 mins | 0.39X |
> > | *Ours (FA3 \+ FA3)* | 27.03 | 0.85 | 2:14 mins | 0.59X | 21.55 | 0.77 | 00:51 | 0.605X |
> > | *Ours \+ SVG2 (FA3+FA3)* | 27.03 | 0.87 | 2:10 mins | 0.45X | 21.61 | 0.77 | 00:58 | 0.44X |
> >
> > |  | HunyuanVideo 720p |  |  |  |
> > | :---- | ----- | :---- | :---- | :---- |
> > |  | **PSNR** | **SSIM** | **Generation Time** | **Attention TFLOPs** |
> > | Dense (FA3) | inf. | 1.0 | 06:56 mins | 1X |
> > | SpargeAttention (FA3 warmup, fp8 inference) | \<20 | \<0.6 | 05:04 mins | \- |
> > | SparseVideoGen2 (FA3+FA3) | 25.80 | 0.87 | 06:24 mins | 0.44X |
> > | *Ours (FA3+FA3)* | 24.02 | 0.80 | 6:30 mins | 0.68X |
> > | *Ours+SVG2 (FA3)* | 24.8 | 0.84 | 6:11 mins | 0.54X |
> >
> > **Experiment Setup:** We generated videos for prompts from the Penguin Benchmark \[G\] (consistent with methodology in prior work \[C\]) for our evaluation. We generated videos corresponding to the first 21 prompts in the benchmark for every video model. For a fair comparison, among all training-free acceleration methods, we set the number of “warmup” steps (initial steps running full attention) to be equal (24% of inference steps). This is consistent with methodology in prior work \[C\].
> >
> > From the table below, we make the following observations:
> >
> > - SpargeAttn can generate videos much faster for every video model because it computes attention at lower precision (SpargeAttn uses 8-bit quantization, whereas SVG2/ours uses bf16), thus, this is not an apples-to-apples comparison. Quantization is orthogonal to both FG-Attn and SVG2, and can be incorporated into these implementations as well. The impact of lower precision can be seen in  HunyuanVideo, where SpargeAttention produces significantly different videos when compared to the baseline, leading to significantly lower PSNR/SSIM.
> > - FG-Attn achieves higher speedups compared to the concurrent state-of-the-art work, Sparse-VideoGen 2 \[C\] (at 16-bit precision), on Wan 14B and Wan 1.3B at 480p resolution. While Sparse-VideoGen 2 outperforms FG-Attn at 720p resolution on HunyuanVideo, Wan 14B, and Wan 1.3B, by a small margin, despite a much larger reduction in FLOPs. This is because Sparse-VideoGen 2 incurs significant overhead from computing query and key clusters, which becomes a larger portion of the total runtime on faster models. However, reordering query tokens into clusters before applying FG-Attn (denoted as **Ours \+ SVG2**) results in performance that exceeds SVG2 across all baselines.
> >
> > We will include these results in the final draft (Section 5.3).
> >
> > **References.**
> > \[A\] Dao, Tri. "Flashattention-2: Faster attention with better parallelism and work partitioning." *arXiv preprint arXiv:2307.08691* (2023).
> > \[B\] Ye, Zihao, et al. "Flashinfer: Efficient and customizable attention engine for llm inference serving." *arXiv preprint arXiv:2501.01005* (2025).
> > \[C\] Yang, Shuo, et al. "Sparse VideoGen2: Accelerate Video Generation with Sparse Attention via Semantic-Aware Permutation." *arXiv preprint arXiv:2505.18875* (2025).
> > \[D\] Spector, Benjamin F., et al. "Thunderkittens: Simple, fast, and adorable ai kernels." *arXiv preprint arXiv:2410.20399* (2024).
> > \[E\] Shah, Jay, et al. "Flashattention-3: Fast and accurate attention with asynchrony and low-precision." *Advances in Neural Information Processing Systems* 37 (2024): 68658-68685.
> > \[F\] Zhang, Jintao, et al. "SpargeAttention: Accurate and Training-free Sparse Attention Accelerating Any Model Inference." *Forty-second International Conference on Machine Learning*.
> > \[G\] Penguin Benchmark: https://huggingface.co/datasets/a-r-r-o-w/penguin-video-benchmark
> > \[H\] P. Zhang et al., "VSA: Faster Video Diffusion with Trainable Sparse Attention," in Advances in Neural Information Processing Systems (NeurIPS), 2025\.

---

### Official Review · Reviewer_Hs2t · 2025-10-29

**Soundness:** 2
**Presentation:** 2
**Contribution:** 2
**Rating:** 2
**Confidence:** 3

**Summary:**

This paper proposes FG-Attn, a fine-grained sparse attention mechanism designed to accelerate diffusion transformers (DiTs) by exploiting slice-level sparsity ($M \times 1$). The authors introduce a custom GPU "gather-load" primitive to efficiently handle the resulting irregular memory access.

**Strengths:**

The primary strength of this work is its ability to apply fine-grained sparsity to accelerate inference without a noticeable degradation in output quality. As demonstrated in Table 2 and the visual examples in the appendix (Figures 16-18), the videos generated using FG-Attn are qualitatively comparable to the baseline.

**Weaknesses:**

1.  **Significant Presentation Issues and Typos:** The paper is in poor shape and is difficult to read.
    * **Lack of Focus:** A large portion of the paper is dedicated to basic, well-known background on diffusion models and standard GPU architecture (e.g., Section 2, Appendix B). This space would be far better utilized by expanding on the core novel contribution—the gather-load primitive and its implementation challenges.
    * **Numerous Typos:** The paper is riddled with simple typographical errors that betray a lack of proofreading. For example:
        * Line 052: "Thus,the" is missing a space.
        * Line 077: "slices of can reduce" is grammatically incorrect.
    * **Incorrect LaTeX Formatting:** Mathematical notation is not formatted to publication standards.
        * Line 137 (and many other places): The `log` function should be typeset as `\log` ($log$ vs. $\log$).
        * Mathematical terms like "data" or "tile" should be properly typeset using `\mathrm` (e.g., $p_{\mathrm{data}}$) for clarity.
        * The LaTeX formatting in Lines 270-274 appears to be broken and is unreadable.

2.  **Limited Novelty:** The core contribution is a hardware-aware algorithm, which the authors acknowledge. However, this contribution appears to be an incremental engineering optimization rather than a fundamental new idea. The work feels like a specialized application of the same principles (IO-awareness, overlapping compute and memory) that made **FlashAttention** successful, but applied to a different sparsity pattern (slices instead of blocks). It doesn't introduce a new conceptual framework for attention or sparsity.

3.  **Limited Performance Gain:** The reported end-to-end speedups (up to 1.65x, with a 1.48x geometric mean) are moderate compared to other methods like SVG. Given the significant, non-trivial engineering effort required to design, implement, and debug a custom CUDA kernel, this performance gain feels limited and may not justify the added complexity over simpler, existing block-sparse methods, though I sincerely appreciate your effort on this.

4.  **Misalignment with ICLR:** This paper's contribution is almost entirely at the systems level. The novelty lies in the CUDA kernel implementation, not in the model or the learning paradigm. A stronger paper for ICLR would **investigate the learning dynamics of DiTs** to understand *why* this fine-grained sparsity emerges and then propose methods to **leverage or induce this sparsity at the model level**. As-is, this work would be a much better fit for a systems-focused conference (e.g., **MLSys**), where a deep dive into the hardware implementation would be the main focus.

**Questions:**

See weakness.

---

> ### Author Response · Authors · 2025-11-28
> **Response to review (1/1)**
>
> Dear reviewer,
>
> Thank you for your feedback and comments\! Please find the answers to your question below.
>
> **LaTeX formatting issues and lack of focus.** We have fixed the formatting and grammatical errors  in the updated version of the draft. Thank you for pointing them out.
>
> **Limited novelty.** The core contribution is an algorithm that leverages both characteristics of video diffusion models and hardware architecture to provide a highly-efficient implementation. Additionally, we make the observation that attention maps in diffusion transformers contain a significant amount of fine-grained sparsity that cannot be easily leveraged by existing block sparse attention implementations for faster inference.
>
> **Limited Performance Gain for the complexity.** We respectfully disagree because similar to other prior works \[F, G, H, I\] which were also published in ML conferences:
>
> 1) Even seemingly small gains in performance for such important and widely used kernels, can lead to significant savings in cost and time. Given that datacenter operational expenditure is forecast to exceed 10s of billions annually \[J\], even a 10-20% efficiency gain will save many billions of dollars. For example, recent work utilizing SparseGPT \[K\] leverages 2:4 sparsity in LLM decoding to achieve a production speedup of approximately 20% end-to-end \[L, M\]. When compounded at the datacenter scale, these efficiency gains translate into massive reductions in operational costs and energy consumption. The importance and difficulty of optimized algorithms and their implementations cannot be understated.
> 2) The engineering complexity is a one-time cost for a given attention module. Once the implementation is available (we will make it open-source), there is no difficulty in adoption for use.
>
> **Misalignment with ICLR.** We would like to highlight that “infrastructure, software libraries, hardware, etc.” *is among the topics mentioned by the ICLR call for papers*. ICLR in the past has accepted many such papers, many of which have had very high impact in the community. Some very recent examples are \[A, B, C, D, E\].
>
> \[A\] R-Topk (ICLR 2025): [https://openreview.net/forum?id=PHg4rAXFVH](https://openreview.net/forum?id=PHg4rAXFVH)
> \[B\] ThunderKittens : [https://openreview.net/forum?id=0fJfVOSUra](https://openreview.net/forum?id=0fJfVOSUra)
> \[C\] FlashMask (ICLR 2025): [https://openreview.net/forum?id=wUtXB43Chi](https://openreview.net/forum?id=wUtXB43Chi)
> \[D\] FlashAttention-2 (ICLR 2024\) [https://openreview.net/forum?id=mZn2Xyh9Ec](https://openreview.net/forum?id=mZn2Xyh9Ec)
> \[E\] Autochunk (ICLR 2024): [https://openreview.net/pdf?id=GQGNLEHmdl](https://openreview.net/pdf?id=GQGNLEHmdl)
> \[F\] S. Yang et al., "Sparse VideoGen 2: Accelerate Video Generation with Sparse Attention via Semantic-Aware Permutation," in *Advances in Neural Information Processing Systems (NeurIPS)*, 2025
> \[G\] J. Zhang et al., "SpargeAttention: Accurate and Training-free Sparse Attention Accelerating Any Model Inference," in International Conference on Machine Learning (ICML), 2025\.
> \[H\] H. Xi et al., "Sparse VideoGen: Accelerating Video Diffusion Transformers with Spatial-Temporal Sparsity," in *International Conference on Machine Learning (ICML)*, 2025\.
> \[I\] X. Li et al., "Radial Attention: O(n log n) Sparse Attention with Energy Decay for Long Video Generation," in Advances in Neural Information Processing Systems (NeurIPS), 2025\.
> \[J\][https://www.forbes.com/sites/tiriasresearch/2023/05/12/generative-ai-breaks-the-data-center-data-center-infrastructure-and-operating-costs-projected-to-increase-to-over-76-billion-by-2028/](https://www.forbes.com/sites/tiriasresearch/2023/05/12/generative-ai-breaks-the-data-center-data-center-infrastructure-and-operating-costs-projected-to-increase-to-over-76-billion-by-2028/)
> \[K\] Frantar, Elias, and Dan Alistarh. "Sparsegpt: Massive language models can be accurately pruned in one-shot." International conference on machine learning. PMLR, 2023\.
> \[L\][https://developers.redhat.com/articles/2025/02/28/24-sparse-llama-smaller-models-efficient-gpu-inference](https://developers.redhat.com/articles/2025/02/28/24-sparse-llama-smaller-models-efficient-gpu-inference)
> \[M\] [https://docs.pytorch.org/tutorials/advanced/semi\_structured\_sparse.html](https://docs.pytorch.org/tutorials/advanced/semi_structured_sparse.html)

---

### Official Review · Reviewer_rzeK · 2025-10-31

**Soundness:** 3
**Presentation:** 3
**Contribution:** 3
**Rating:** 8
**Confidence:** 4

**Summary:**

This paper presents FG-Attn, a fine-grained sparse attention mechanism designed to accelerate diffusion transformers (DiTs) used for realistic video and image generation. Unlike prior block-sparse approaches that operate on coarse M×M tiles (e.g., 64×64), FG-Attn exploits fine-grained sparsity at the level of M×1 slices, effectively reducing redundant attention computations. The authors address the key challenges of irregular memory access and low tensor core utilization by introducing a novel asynchronous gather-load primitive that efficiently loads only the required key/value vectors into on-chip shared memory. Experiments on state-of-the-art video models show up to 1.65× speedup (1.48× on average) on H100 GPUs, demonstrating that FG-Attn can surpass existing block-sparse attention methods in diffusion transformers.

**Strengths:**

- this paper is well-structured and clearly presented paper.

- this paper addresses an important and timely topic — improving efficiency of diffusion-based video generation models, where attention cost dominates latency.

- this paper has technically deep contribution, including a fine-grained (128×1) sparse attention kernel implementation that pushes the boundary of GPU efficiency on sparse attention.

- the proposed system–algorithm co-design is well thought out, particularly in reducing sparse-index mask generation overhead by leveraging similarity of attention maps between diffusion steps.

**Weaknesses:**

- workload balancing across different query blocks is unclear — fine-grained sparsity may lead to uneven computation workloads, potentially limiting scalability.

- attention mask caching strategy might not generalize well to few-step diffusion settings, where attention patterns can vary more significantly between steps.

**Questions:**

- How does FG-Attn handle load balancing when different query blocks exhibit varying levels of sparsity?

- The paper mentions attention map similarity between diffusion steps to reduce mask generation cost. How robust is this caching strategy in few-step diffusion settings, where the similarity may drop?

---

> ### Author Response · Authors · 2025-11-28
> **Response to review (1/1)**
>
> Dear reviewer,
>
> Thank you for reviewing our paper\! We deeply value your feedback and comments. We address your questions below.
>
> **Load balancing due to different numbers of KVs per query block.** FG-Attn introduces a potential load imbalance: specific query groups require attending to more key-values than others, creating uneven work distribution across thread blocks (and thus, SMs). We find that the load imbalance exhibited by different query blocks within one single head is limited. For example, in the Wan 1.3B 480p model running FG-Attn, the maximum difference in the number of KVs to process within a single attention head as a % of sequence length is at most 18%. The extra latency due to this load imbalance can be hidden by other thread blocks. Thus, we find that load imbalance in FG-Attn is not a major contributing factor to the overhead in the case of DiT models.
>
> **Results on few-step diffusion models.** We acknowledge that FG-Attn would achieve smaller speedup on few-step diffusion models. This is a limitation shared by prior work as well \[A, B, C\]. While FG-Attn is intended to accelerate diffusion models performing inference over a larger number of steps, applying FG-Attn to few-step models still leads to a small speedup. We provide here additional results on implementing FG-Attn on few-step distillation models. We chose a 4-step distilled Wan 2.2 model \[D\] for evaluation.
>
> |  | PSNR | SSIM | Average Generation rate | Generation rate (Original) |
> | :---- | :---- | :---- | :---- | :---- |
> | *FastVideo/FastWan2.2-TI2V-5B  (4 step)* | 24.9 | 0.77 | 1.35 s/it | 1.49s/it |
>
> **References**
> \[A\] S. Yang et al., "Sparse VideoGen 2: Accelerate Video Generation with Sparse Attention via Semantic-Aware Permutation," in *Advances in Neural Information Processing Systems (NeurIPS)*, 2025
> \[B\] X. Li et al., "Radial Attention: O(n log n) Sparse Attention with Energy Decay for Long Video Generation," in Advances in Neural Information Processing Systems (NeurIPS), 2025\.
> \[C\] H. Xi et al., "Sparse VideoGen: Accelerating Video Diffusion Transformers with Spatial-Temporal Sparsity," in *International Conference on Machine Learning (ICML)*, 2025\.
> \[D\] FastVideo Wan 2.2 [https://huggingface.co/FastVideo/FastWan2.2-TI2V-5B-FullAttn-Diffusers](https://huggingface.co/FastVideo/FastWan2.2-TI2V-5B-FullAttn-Diffusers)

---

### Official Review · Reviewer_Kiqv · 2025-11-02

**Soundness:** 2
**Presentation:** 2
**Contribution:** 3
**Rating:** 4
**Confidence:** 4

**Summary:**

The paper introduces FG-Attn, a fine-grained sparse attention mechanism designed to accelerate video diffusion transformers. Existing methods rely on block-sparse attention, skipping coarse-grained tiles, which leaves much of the redundancy unexploited. FG-Attn exploits finer sparsity at the slice level (M×1) by identifying and skipping negligible query-key computations. The approach introduces a hardware-efficient asynchronous gather-load primitive that allows irregular memory access without reducing tensor core utilization, and two training-free mask prediction strategies to identify redundant slices efficiently. Experiments on large video diffusion models such as Wan 2.1 and HunyuanVideo demonstrate speedup in generation time.

**Strengths:**

1. The authors propose a novel fine-grained sparse attention approach that meaningfully extends beyond block-sparse paradigms.

2. Introduces practical GPU-level optimizations (asynchronous gather-load) that effectively hide irregular memory access overheads and implement the algorithm using ThunderKitten, which is valuable.

**Weaknesses:**

My major concern is about evaluation. In Figure 11, the reported speedup number of SVG and RadialAttention is much lower than the number the original speedup number they reported in their paper. Meanwhile, Wan 2.1 720p and Hunyuan 720p offer similar speedup results, which is not consistent with much of previous literature.

Secondly, the authors should report PSNR or SSIM, as they are more reliable benchmarks than VBench (Table 2).

Thirdly, the paper should be specific about whether they are comparing with the FlashAttention-3 baseline or with the FlashAttention-2 baseline. Its known that ThunderKitten is able to produce high-performing kernel that performs on par with FlashAttention-3, therefore the authors need to report the real TFLOPs number in Figure 12 for more reliable comparison.

**Questions:**

Please refer to the weakness section.

---

> ### Author Response · Authors · 2025-11-28
> **Response to review (1/3)**
>
> Dear reviewer,
>
> Thank you for your review\! We appreciate the insightful comments and feedback. Please find the answers to your question below.
>
> **Explaining difference in speedup results for SVG and Radial Attention.** The speedup numbers reported in our paper and SVG/Radial attention differ because:
>
> 1) Flash Attention Baseline: All speedups we initially reported in Figure 11 (prior to revision), are with respect to a Flash Attention 3 baseline (that we implemented using ThunderKittens\[D\]). Radial Attention and SVG calculate speedups with respect to a Flash Attention 2 \[A\] baseline in their papers.
> 2) Radial Attention considers a video length of 129 frames for Hunyuan video, and 69 frames for Wan-14B. We generate an 81 frame video in all of our experiments (similar to SVG2 \[C\]).
>
> We thus clarify that our reported results are accurate. To make this clearer in the submission, we provide the detailed experimental configuration with baselines and the wall-clock time (un-normalized) to generate the videos and their corresponding speedups in the additional results below and in the updated draft (Section 5.3).
>
> ***PSNR, SSIM results:*** We report all PSNR/SSIM results compared to prior work in the table below and in the updated draft (Section 5.3).
>
> **FlashAttention Baseline:**  In the submitted version, the primary baseline is our ThunderKittens \[D\] reimplementation of Flash Attention 3 (FA3). Our sparse attention mechanism was built over this ThunderKittens \[D\] version of FA3.
>
> While earlier attention mechanisms’ implementations such as Radial-Attention and SVG were only available with Flash Attention 2 (FA2), recent works like SVG2 use the open-source Flash-Infer implementation of FA3 \[B\] for their sparse attention mechanism. In the revised version, we present additional comparison with two prior/concurrent works: SpargeAttn\[F\] and Sparse-VideoGen-2\[C\] (Sparse-VideoGen 2 is a very recent concurrent work, unpublished work at the time of submission. The code was also unavailable at the time of submission). To ensure a fair and relevant comparison across all prior work, we re-collected all results (including ours) below with two baselines: one set of results that use a FA2 \[A, B\] baseline (Radial Attention, Sparse-VideoGen), and another set of results using a FA3 baseline \[E\].
>
> We reimplement FG-Attn over FA3 \[E\] (instead of Thunderkittens’ implementation) to compare against the FA3 baseline results, and provide an extended evaluation for different Flash Attention baselines for fair comparisons. (Note that FA3 \[E\] has lower overheads when compared to the ThunderKittens version). Please see the “additional experiments” section below for a detailed experimental setup. We then report the unnormalized wall-clock times and the reduction in attention FLOPs for each mechanism. We have also updated the draft to include these (Section 5.3 of the paper).

---

> > ### Author Response · Authors · 2025-11-28
> > **Response to review (2/3)**
> >
> > ## Additional experiments
> >
> > We have provided here a comparison of the unnormalized wall-clock time, FLOPS, SSIM, and PSNR, between various implementations. We generated videos for prompts from the Penguin benchmark \[G\] (consistent with methodology in prior work \[C\]) for our evaluation. We generated videos corresponding to the first 21 prompts in the benchmark for every video model. For a fair comparison, among all training-free acceleration methods, we set the number of “warmup” steps (initial steps running full attention) to be equal (24% of inference steps). This is consistent with methodology in prior work \[C\]. We note that for some open-source prior works, the baseline FA is different for warmup versus inference (e.g., in SVG2). Thus, we specify for each configuration which FA version is used for the two phases. We list the experiment configurations as follows:
> >
> > 1\. Flash Attention 2 baselines:
> >
> > - Dense (FA2) \- Uses FA2 baseline \[A\]
> > - Radial attention (FA2+FA2): Radial attention with the warmup steps implemented in FA2 \[A\], and the sparse attention implemented over FlashInfer’s FA2 \[B\]
> > - SparseVideoGen2 (FA2+FA3): SVG2 with warmup steps implemented using FA2 and the SVG2 kernel implemented on FlashInfer’s FA3 \[B\]. This is the open-source version of SVG2 available.
> > - SparseVideoGen (FA2 \+ FA2): SVG with the warmup steps being implemented using FA2, and the sparse attention implemented over FlashInfer’s FA2 \[B\]
> > - Ours (FA2+TK-FA3): Our attention mechanism with the warmup steps implemented with FA2, implemented over Thunderkittens FA3 \[D\].
> >
> > We note that some prior works like SVG2 use the open-source FlashInfer implementation of FA3 \[E\], so we recollected all comparisons (including ours) below with the same FA3 \[E\] baseline (FA3) for fair comparison. We have reimplemented FG-Attn over FA3 \[E\] and used this new implementation in this evaluation. We have also updated the draft to include these results (Section 5.3).
> >
> > Flash Attention 3 baselines:
> >
> > - Dense (FA3): Using FA3 \[E\].
> > - SpargeAttention (FA3+fp8): Warmup implemented using FA3 \+ Sparge Attention uses fp8 attention.
> > - SparseVideoGen2 (FA3+FA3): Warmup implemented using FA3 \+ SVG2 on top of FA3\[E\] and the SVG2 kernel implemented on FlashInfer’s FA3 \[B\].
> > - Ours (FA3+FA3): Our attention mechanism with the warmup steps implemented and sparse attention kernel implemented over FA3 \[E\]
> > - Ours \+ SVG2 (FA3+FA3): FG-Attn mechanism built on top of FA3 \[E\]. Before each attention run, the queries are permuted using the SVG2 clustering strategy.
> >
> >
> > We present the results in the tables below, we make the following observations:
> >
> > - On Wan-14B 720p, Wan 1.3B 720p and 480p and HunyuanVideo, while FG-Attn is slower, it outperforms SVG and Radial Attention in terms of video quality in the produced output video. The slowdown is because the ThunderKittens implementation of FA3 incurs higher overhead for longer sequences. The FA3 implementation of FG-Attn outperforms both mechanisms.
> > - SpargeAttn can generate videos much faster for every video model because it computes attention at lower precision (SpargeAttn uses 8-bit quantization, whereas SVG2/ours uses bf16), thus, this is not an apples-to-apples comparison. Quantization is orthogonal to both FG-Attn and SVG2, and can be incorporated into these implementations as well. The impact of lower precision can be seen in  HunyuanVideo, where SpargeAttention produces significantly different videos when compared to the baseline, leading to significantly lower PSNR/SSIM.
> > - FG-Attn achieves higher speedups compared to the concurrent state-of-the-art work, Sparse-VideoGen 2 \[C\] (at 16-bit precision), on Wan 14B and Wan 1.3B at 480p resolution. While Sparse-VideoGen 2 outperforms FG-Attn at 720p resolution on HunyuanVideo, Wan 14B, and Wan 1.3B, by a small margin, despite a much larger reduction in FLOPs. This is because Sparse-VideoGen 2 incurs significant overhead from computing query and key clusters, which becomes a larger portion of the total runtime on faster models. However, reordering query tokens into clusters before applying FG-Attn (denoted as **Ours \+ SVG2**) results in performance that exceeds SVG2 across all baselines.

---

> > > ### Author Response · Authors · 2025-11-28
> > > **Response to review (3/3)**
> > >
> > > |  | Wan 14B 720p |  |  |  | Wan 14B 480p |  |  |  |
> > > | :---- | ----- | :---- | :---- | :---- | ----- | :---- | :---- | :---- |
> > > |  | **PSNR** | **SSIM** | **Generation Time (mins)** | **Attn TFLOPs** | **PSNR** | **SSIM** | **Generation Time (mins)** | **Attn FLOPs** |
> > > | Dense (FA2) | inf. | 1.0 | \~28 | 1X | inf. | 1.0 | 7:30 | 1X |
> > > | RadialAttention (FA2+FA2) | 21.1 | 0.72 | 20:45 | 0.48X | 22.01 | 0.76 | 6:24 | 0.48 X |
> > > | SVG2 (FA2+FA3) | 22.39 | 0.78 | 17:17 | 0.47X | 23.48 | 0.84 | 6:05 | 0.47X |
> > > | SVG (FA2 \+ FA2) | 19.82 | 0.70 | 17:32 | 0.47X | 20.13 | 0.74 | 5:18 | 0.47X |
> > > | *Ours (FA2+TK-FA3)* | 23.01 | 0.79 | 18:29 | 0.68X | 22.04 | 0.78 | 5:41 | 0.68X |
> > > | Dense (FA3) | inf. | 1.0 | 16:45 | 1X | inf. | 1.0 | 4:12 | 1X |
> > > | SpargeAttention (FA3 warmup, fp8 inference) | 22.51 | 0.79 | 13:04 | \- | 20.11 | 0.78 | 3:58 | \- |
> > > | SVG2 (FA3+ FA3) | 22.39 | 0.78 | 15:25 | 0.47X | 23.48 | 0.84 | 4:58 | 0.47X |
> > > | *Ours (FA3+FA3)* | 23.01 | 0.79 | 15:55 | 0.68X | 22.04 | 0.78 | 4:02 | 0.68X |
> > > | *Ours \+ SVG2 (FA3)* | 22.93 | 0.78 | 15:14 | 0.53X | 22.31 | 0.78 | 4:01 | 0.53X |
> > >
> > > |  | Wan 1.3B 720p |  |  |  | Wan 1.3B 480p |  |  |  |
> > > | :---- | ----- | :---- | :---- | :---- | ----- | :---- | :---- | :---- |
> > > |  | **PSNR** | **SSIM** | **Generation Time (mins)** | **TFLOPs** | **PSNR** | **SSIM** | **Generation Time (mins)** | **Attn FLOPs** |
> > > | *Dense (FA2)* | inf. | 1.0 | 06:46 | 1X | inf. | 1.0 | 1:38  | 1X |
> > > | SVG2 (FA2+FA3) | 27.60 | 0.87 | 3:22 | 0.39X | 22.44 | 0.82 | 1:09  | 0.39X |
> > > | SVG (FA2+FA2) | 22.40 | 0.78 | 4:01 | 0.47X | 16.70 | 0.64 | 1:07  | 0.47X |
> > > | *Ours (FA2 \+ TKFA3)* | 27.03 | 0.85 | 4:09 | 0.59X | 21.55 | 0.77 | 1:08  | 0.605X |
> > > | Dense (FA3) | inf. | 1.0 | 2:53 | 1X | inf. | 1.0 | 1:01  | 1X |
> > > | Sparge (FA3 warmup, fp8 inference) | 27.10 | 0.84 | 2:26 | \- | 21.41 | 0.79 | 00:43  | \- |
> > > | SVG2 (FA3 \+ FA3) | 27.60 | 0.87 | 2:30 | 0.39X | 22.44 | 0.82 | 1:00  | 0.39X |
> > > | *Ours (FA3 \+ FA3)* | 27.03 | 0.85 | 2:14 | 0.59X | 21.55 | 0.77 | 00:51 | 0.605X |
> > > | *Ours \+ SVG2 (FA3+FA3)* | 27.03 | 0.87 | 2:10 | 0.45X | 21.61 | 0.77 | 00:58 | 0.44X |
> > >
> > > |  | HunyuanVideo 720p |  |  |  |
> > > | :---- | ----- | :---- | :---- | :---- |
> > > |  | **PSNR** | **SSIM** | **Generation Time (mins)** | **Attn FLOPs** |
> > > | *Dense (FA2)* | inf. | 1.0 | 12:38 | 1X |
> > > | RadialAttention (FA2) | 22.33 | 0.77 | 09:07 | 0.48 X |
> > > | SVG2 (FA2 \+ FA3) | 25.85 | 0.87 | 06:31 | 0.44X |
> > > | SVG (FA2) | 20.19 | 0.76 | 06:01 | 0.45X |
> > > | *Ours (FA2+TK-FA3)* | 24.02 | 0.80 | 07:32 | 0.68X |
> > > | Dense (FA3) | inf. | 1.0 | 06:56 | 1X |
> > > | SpargeAttention (FA3 warmup, f8 inference) | \<20 | \<0.6 | 05:04 | \- |
> > > | SVG2+FA3 | 25.80 | 0.87 | 06:24 | 0.44X |
> > > | *Ours (FA3)* | 24.02 | 0.80 | 06:30 | 0.68X |
> > > | *Ours \+SVG2 (FA3)* | 24.8 | 0.85 | 06:10 | 0.54X |
> > >
> > >
> > >
> > >
> > >
> > > ***References***
> > >
> > > \[A\] Dao, Tri. "Flashattention-2: Faster attention with better parallelism and work partitioning." *arXiv preprint arXiv:2307.08691* (2023).
> > > \[B\] Ye, Zihao, et al. "Flashinfer: Efficient and customizable attention engine for llm inference serving." *arXiv preprint arXiv:2501.01005* (2025).
> > > \[C\] Yang, Shuo, et al. "Sparse VideoGen2: Accelerate Video Generation with Sparse Attention via Semantic-Aware Permutation." *arXiv preprint arXiv:2505.18875* (2025).
> > > \[D\] Spector, Benjamin F., et al. "Thunderkittens: Simple, fast, and adorable ai kernels." *arXiv preprint arXiv:2410.20399* (2024).
> > > \[E\] Shah, Jay, et al. "Flashattention-3: Fast and accurate attention with asynchrony and low-precision." *Advances in Neural Information Processing Systems* 37 (2024): 68658-68685.
> > > \[F\] Zhang, Jintao, et al. "SpargeAttention: Accurate and Training-free Sparse Attention Accelerating Any Model Inference." *Forty-second International Conference on Machine Learning*.
> > > \[G\] Penguin Benchmark: https://huggingface.co/datasets/a-r-r-o-w/penguin-video-benchmark

---

### Author Response · Authors · 2025-12-04
**Official Comment by Authors**

We thank the reviewers for their insightful comments. We have addressed all the questions and concerns raised by the reviewers and made the following changes to our draft:

* **Expanded Results Section (Section 5.3):** We updated the draft to include comparisons against Radial Attention, SpargeAttn, and the concurrent Sparse-VideoGen2 (SVG2). We provided these comparisons across both FlashAttention-2 and FlashAttention-3 baselines.
  * We conclude that for the same generation quality, FG-Attn performs better than all prior works (FlashAttention-3 baseline).
  * Compared to *concurrent* work SVG2, we perform comparably (FG-Attn performs better on 480p videos \- a shorter sequence of tokens \- while performing comparably to SVG2 on 720p). However, this approach is orthogonal *and can be combined to generate the currently fastest attention kernels for video generation* for all configurations. We evaluate this under **Ours \+ SVG2:**  a configuration that combines the sparse attention kernel of FG-Attn with the token reordering strategy of Sparse-VideoGen 2\.
* **Quality Metrics:** We now report un-normalized wall-clock times, Attention FLOPs, PSNR, and SSIM for all experiments (based on feedback from Reviewers Kiqv and Esgn). This allows for a direct comparison of absolute generation speed and video quality. **"**
* **Clarifying novelty and impact:** We highlight that our primary contribution is:
  * Identifying untapped fine-grain sparsity in the attention masks of video diffusion models
  * Leveraging this fine-grained sparsity specific to video diffusion models, which existing block-sparse methods cannot easily leverage with novel hardware-optimized techniques.
  * Our approach currently provides state-of-art speedup for DiT attention kernels for video generation compared to all prior works and the currently fastest available attention kernels for video generation in conjunction with concurrent work SVG2.
  * We further highlight that the engineering complexity is a one-time cost yielding efficiency gains that, while seemingly small, translate to massive operational savings at datacenter scale.
* **Formatting error fixes:** We corrected all LaTeX formatting and grammatical errors throughout the paper.

---

### Note · Program_Chairs · 2026-01-17
**Submission Desk Rejected by Program Chairs**

The following references in this submission do not refer to real documents and/or have major errors in bibliographic information:

 Patrick Esser, Michael S. M. Townsend, Sumith Kulal, Tim Dockhorn, Jonas Müller, Anastasiia Alterovych, David Dehaerne, Peter T. H. Lu, Caner Hazirbas, Dominic Rampas, Robin Rombach, Joachim D’Asaro, Daniel Watson, Daniel Voinea, Liezl Puzon, Y-Lan Boureau, and Fabian Mentzer.